# Secure IIoT Information Reinforcement Model Based on IIoT Information Platform Using Blockchain

**DOI:** 10.3390/s22124645

**Published:** 2022-06-20

**Authors:** Yoon-Su Jeong

**Affiliations:** Department of Information Communication Engineering, Mokwon University, Daejeon-si 35349, Korea; bukmunro@mokwon.ac.kr; Tel.: +82-42-829-7678

**Keywords:** blockchain, IIoT, big data, platform, reinforcement, information management technology

## Abstract

Data created at industrial sites through industrial internet of things devices are now being processed automatically or in real-time in the industrial structure, due to the application of artificial intelligence technology to industrial sites. However, the expenses of autonomous or real-time data processing and steady data processing (analysis, prediction, prescription, and implementation) necessitate a new processing method. We propose a blockchain-based industrial internet of things information reinforcement model in this work that may reliably ensure the integrity of industrial internet of things data produced at industrial locations. The proposed model processes industrial internet of things data that may occur at endpoints at industrial sites into the blockchain by processing data generated by the same industrial internet of things device independently. As a result, the IIoT data sent to the industrial internet of things server can be evaluated more readily, and production accuracy may be enhanced. The proposed model optimizes industrial internet of things information linkage by stochastically reflecting the information based on attribute value frequency. By dynamically aggregating the related data of industrial internet of things information acquired as a seed through hierarchical subnets, the proposed model increases stability and accuracy. Furthermore, the proposed model may be used to enhance an organizations’ operational efficiency (consulting and training, for example) and strategic decision-making by utilizing fundamental knowledge about items produced at industrial locations. Furthermore, the proposed model allows for information sharing and system connectivity between industrial locations, allowing for close collaboration between industrial internet of things features. As a result of the performance evaluation, the proposed model included an industrial internet of things sensor to the blockchain, eliminating the need for an extra function in the manufacturing process and reducing the time required to validate the integrity of industrial internet of things data. In addition, as a result of analyzing industrial internet of things data by an algorithm according to the number of simulated clouds, the accuracy of industrial internet of things information was improved by 2.5% to 3%, on average.

## 1. Introduction

The industrial internet of things (IIoT) which combines low-power processes and network cards, has recently been applied at production sites to efficiently produce and evaluate product information across networks in numerous industries [1]. The industrial internet of things (IIoT) is a subset of the internet of things (IoT), which connects sensors and networks in many industrial domains. The influence of IIoT on enterprises and daily life is greater than that of IoT. The most significant distinction between IIoT and IoT is that IIoT connects industrial machinery and gadgets. IIoT follows the same principles as IoT, but it adds automated layers and reporting capabilities. However, due to a lack of standards, integration with legacy technologies, pre-investment expenditures, and knowledge, it is not effectively employed in the industrial context [2].

Existing IIoT studies look for ways to collect and analyze IIoT datasets to lower production costs while producing high-quality products at industrial sites [3,4]. However, it is critical to managing IIoT wisely since IIoT in industrial settings is utilized in many ways depending on the objective. It is also necessary to secure the integrity of the data collected through IIoT. The amount of research being performed on setting up a testbed that transmits and receives signals using IIoT sensors in industrial settings is rapidly expanding. In particular, testbeds are being constructed in industrial locations to be reusable in non-specific situations and specialized scenarios, depending on the type of data collected and the type of analysis [5].

Furthermore, blockchain and smart contracts are being used in several studies on data sharing access control in IIoT networks, such as resource and data access control between IoT devices. However, there are some challenges when blockchain is used directly to IIoT data-sharing networks. Due to the inability of most low-power IIoT devices to participate in the consensus process, IIoT data exchange must incorporate data storage. Furthermore, frequent transmission of shared data requires more energy in low-power IoT devices, and large-scale access may create severe interference. Edge computing and consortium blocks, as well as IIoT devices and industrial interaction patterns, must be incorporated into IIoT data-sharing networks. To date, research in the industrial sector has been insufficient to handle the problem of data sharing and secure the integrity of IIoT information while dispersing data sharing by integrating the produced information between IIoT devices with blockchain.

We propose an IIoT information reinforcement model based on the IIoT information platform that uses blockchain to enhance IIoT data generated at industrial sites in this study. The proposed model allows for strengthening IIoT information integrity and the close linking of diverse IIoT information to improve IIoT information while lowering costs in small manufacturing enterprises. The proposed model maintains the integrity of IIoT data acquired at industrial sites and allows for information uniqueness and system connectivity inside industrial sites, allowing for close collaboration between IIoTs with various features. 

The proposed model generates a subnet by stochastically applying the frequency of attribute information of the acquired information to deep learning in order to collect and analyze IIoT data efficiently. Using inter-subnet connection information, different subnets are grouped in a hierarchical framework. This is to improve the IoT device’s dynamic clustering speed and accuracy. To evaluate IIoT data efficiently, the proposed model used a deep learning technique. Deep learning can be employed in the proposed model to accurately check IIoT information, ensuring the data integrity of IIoT devices at a cheap cost.

Compared with the typical IoT processing model employed in existing industrial locations, the suggested model has the following properties. The proposed model may collect real-time product data from IIoT sensors at industrial locations and use it in the product development process. Second, production product information, such as faults that may arise in industrial sites, can be tracked and monitored in real-time. Third, the integrity of the information collected from the IIoT sensor may be easily verified. Fourth, managers may easily control IIoT devices from afar because it runs on the cloud. Fifth, the linkage plan with the existing system is smoother than the existing manufacturing process while increasing the production environment.

The remainder of this study is organized as follows: Section 2 looks at the definition of IIoT and related research, Section 3 presents an IoT information reinforcement model based on the IIoT information platform and blockchain, and Section 4 evaluates the proposed model’s performance through simulation. Finally, Section 5 brings us to a close.

## 2. Background

### 2.1. Smart Factory vs. Factory Automation

Recently, most of the items produced at the industrial facility have been created by machines. Companies fabricated most of their products by hand in the past. However, in recent years, they have switched to semi-automated or automated methods to cut management and production costs and enhance management performance [6]. Companies have used industrial automation and smart factories interchangeably until recently. However, depending on the socio-structural changes in the manufacturing business, safeguarding competitiveness and productivity, and enhancing efficiency, factory automation or smart factories have dramatically varied meanings [7]. The ideas of industrial automation and smart factory are compared and defined in Table 1.

Factory automation is a production system mechanism that manages the functioning of a machine without relying on people, and it refers to an unmanned factory that automates the majority of processes. Factory automation is used in the disciplines of computerized design, computerized production, and integrated manufacturing and typically comprises control systems such as conveyors, automatic warehouses, numerical control machine tools, and quality inspection devices [8,9].

The ideas of industrial automation and smart factory are compared and defined in Table 1.

Factory automation is a production system mechanism that manages the functioning of a machine without relying on people, and it refers to an unmanned factory that automates the majority of processes. Factory automation is used in the disciplines of computerized design, computerized production, and integrated manufacturing and typically comprises control systems such as conveyors.

### 2.2. Blockchain System for IIoT

IIoT blockchain systems are created and utilized in industrial locations, combining IoT with blockchain. They offer IIoT blockchain system procedures trust building, cost reduction, data trading market activation, and increased security [10]. As illustrated in Figure 1 [11,12], the IIoT blockchain system, a blockchain-based internet of things technology, is employed by splitting the blockchain and IoT platform’s interworking process into an interworking process and a direct interworking process.

The connection between the blockchain and the internet of things, as depicted in Figure 1, applies the acquired data to the blockchain network. The connection of the blockchain and IoT platform has the benefit of allowing current IoT to coexist by translating data into blockchain form via blockchain proxy. It also has the disadvantage of relying on the IoT platform.

IIoT blockchain solutions are primarily used for authentication, fraud prevention, and the creation of a shared environment [6]. Aiota, Streamer, and Japan’s Nayuta are good instances of IIoT blockchain creation and functioning. The IIoT blockchain system may be utilized in the distribution/logistics industry, but it must compensate for the centralized method’s drawbacks of poor speed, inefficiency, and dispersed ledger.

Permissionless Proof of Work (PoW) systems (e.g., Bitcoin and Ethereum 1.0) enable scalability, distributed processing, and security, while most existing blockchain systems do not. Furthermore, centralized block production systems (e.g., Cardano, EOS) aim for scalability at the price of block producer variance. Meanwhile, multi-chain systems (such as Cosmos and AION) achieve scalability, dispersibility, and high-speed TTF at the risk of further assaults.

## 3. Related Works

IIoT technologies have recently advanced significantly in various industries (environment, agricultural, monitoring, surveillance, etc.) [13]. Blockchain technology, in particular, can process, store, and distribute data generated from IIoT devices, addressing some of IIoT’s shortcomings (security concerns, high operational costs, frequency delay, etc.) [14]. 

Miller et al., showed how to integrate IIoT with blockchain to boost productivity in many businesses [15]. Liang et al., suggested a communication architecture for IIoT applications [16]. This technology can operate IIoT applications’ data assurances, robustness, and accountability flexibly. To maintain the security and privacy of smart grid energy data consumption, Aitzhan et al., deployed multi-coded and anonymous encrypted communications streams [17]. Tesla and colleagues proposed a blockchain-based IoT platform [18]. However, there is a difficulty in that the blockchain system’s performance (expandability, decentralization, security, or latency) is not well-specified.

Keping et al., developed a blockchain reinforcement complementary solution for smart industrial data to safe storage, access control, information update and deletion, and tracking and termination of bad users [19]. This solution needs the system to function under the Decision Bilinear Diffie–Hellman (DBDH) assumption to monitor or cancel rogue users at all stages.

Quansi et al., presented a blockchain-based data-sharing approach for merging IIoT device monitoring and recording with smart contract storage on the network [20]. Only firms who meet the access policy to intelligent contracts may execute and examine transaction data since this approach provides a collaborative solution for IIoT devices.

Aparna et al., presented a distributed approach based on blockchain that focused on the challenge of data distribution in IIoT systems [21]. Assuming that the P2P network is safe, this paradigm allows any node to interact with another node. However, via case studies of intelligent grid systems for IIoT, the model employed data load balancing, energy management costs, and transfer delay characteristics to deliver various services.

Aamir et al., made a comprehensive framework for the intelligent industry, encompassing the significant concerns and challenges of IIoT and blockchain IIoT [22]. Converging with blockchain can yield several benefits. However, before IoT adoption, the government and other policies must address general layouts.

Shen et al., proposed a trustworthy blockchain-based cooperation model that includes data owners, minors, and third parties [23]. This solution minimizes the income distribution process by gathering and storing shared data over a private or public cloud. However, because data owners, minors, and third parties all share a blockchain, this approach has a flaw: everything is documented by intelligent contracts.

Liu et al., developed a data-sharing strategy based on the Ethereum blockchain and deep reinforcement learning (DRL) [24]. The approach collects the most quantity of data possible while also ensuring data security and dependability. It also has greater security and resilience to assaults than traditional database-based data exchange methods.

Using blockchain technology, Shen et al., proposed a safe and trustworthy shared platform across many data sources [25]. Paillier, a homomorphic cryptographic system, is used on the platform, which uses safe building blocks such as polynomial multiplication and security comparisons. Furthermore, the platform protects each data provider’s vital data and the SVM model parameters.

Wang et al., proposed a framework that uses Ethereum blockchain and ABE technology to integrate data storage and sharing strategies in distributed storage systems [26]. This framework, based on smart contracts, performs a forward search function on password text in distributed storage systems, addressing the issue of not returning all or inaccurate results.

On mobile cloud platforms, Nguyen et al., proposed a shared architecture integrating blockchain and the interplanetary file system (IPFS) [27]. This framework provides a dependable data-sharing solution in the mobile cloud while protecting sensitive health data from potential attacks.

Sun et al., proposed an attribute-based encryption strategy based on cryptographic policy attribute-based encryption systems paired with blockchain for securely storing and distributing electronic medical records in an IPFS storage environment [28]. The solution prevents a single point of failure by keeping encrypted electronic medical data in a distributed interplanetary file system (IPFS).

Table 2 examined the contents of the previous research by categorizing them as related works, related techniques, advantages, and limitations.

## 4. An IoT Information Reinforcement Model Using Blockchain

In addition to data collecting, IIoT sensors used in industrial locations strive to handle IIoT sensing information securely and effectively while minimizing network burden. In this study, we propose a blockchain-based IIoT information reinforcement model to ensure the integrity of IIoT sensing information and facilitate information uniqueness and system connection in industrial settings. The proposed model provides data processing speed, accuracy, and stability by hierarchically dividing IIoT sensing information into subnets and dynamically grouping the related information of seeds amongst the built subnets. In addition, the proposed model performs interactive communication between IIoT and servers, cross-verifying IIoT sensing information in a time series, providing IIoT information weight of certain probability information. The reason is that the time required is shortened by comparing and analyzing the impact on performance by selecting various types of IIoT information according to the characteristics and types of IIoT information in dimensionally reducing IIoT information.

### 4.1. Overview

Small industrial sites recently have produced unrecognizable production data among the acquired data, so they tend to collect data from a portion of the production process rather than the full industrial site. Furthermore, because the development cycle of development goods is quick in domains such as R&D, it is difficult to produce appropriate analysis findings for data collected at the production site. It is vital to make the most of information collection, analysis, and exchange to properly fuse and integrate large-scale production data created at industrial locations.

To advance IIoT information generated at industrial sites, we propose a reinforcement model based on the IIoT information platform utilizing blockchain in this study. The proposed model allows for strengthening IIoT information integrity and the close linking of diverse IIoT information to improve IIoT information while lowering costs in small manufacturing enterprises. The proposed model gathers IIoT data from industrial locations in real-time, independent of time or location. Because a considerable amount of IIoT sensing data was created in this situation, a probability-based deep learning clustering approach was used to identify the IIoT sensing data effectively. Information sharing and system linkage inside industrial sites are made easier due to this procedure, and intimate connections between IIoT with varied features are conceivable.

The proposed model generates a subnet by stochastically applying the frequency of attribute information of the acquired information to deep learning in order to collect and analyze IIoT data efficiently. Different subnets are aggregated into hierarchical structures utilizing inter-subnet connection information to increase the dynamic clustering speed and accuracy of IIoT information. The proposed model uses deep learning techniques to evaluate IIoT data effectively, enhancing accuracy and ensuring the data integrity of IIoT devices at a cheap cost. 

### 4.2. Information Gathering for IIoT

The proposed model splits the process of gathering IIoT data into two sections. The first is a transaction area, which stores and processes data from IIoT devices, and the second is a blockchain system area, which processes data at industrial locations. 

Figure 2 shows how the blockchain-based IIoT method, used in industrial settings, saves money on production efficiency, labor, and production expenses. For the IIoT process, such as Figure 2, to be applied to an industrial site, current industrial site facilities must be changed, which creates an issue in terms of additional facility replacement costs. Customized services at industrial locations are conceivable if the IIoT process is moved to a blockchain, as described in the model. The information acquired in IIoT must be continually added as a block in the manner depicted in Figure 2. This is because freshly created blocks combine with existing blocks to form a local blockchain.

Figure 3 depicts the data collection model’s procedure for gathering various data from IIoT sensors deployed at industrial locations in the proposed model. As shown in Figure 3, data is collected from sensors put in various manufacturing equipment developed at industrial locations, and production product data is modeled and serviced by reflecting needs.

As indicated in Figure 3, the following conditions must be addressed in order for IIoT data gathering to be processed securely and efficiently at industrial locations. First, connections should allow current facilities and equipment on industrial sites to be replaced or upgraded. Second, protocols and communication interfaces between diverse systems must be developed. Third, IIoT data should be sent and received across a network, such as a cable, wireless, RFID, or Wi-Fi. Fourth, equipment constructed at industrial locations should be able to be relocated and used. Fifth, data sent and received between devices should be safeguarded via a secure channel. By applying the findings of data analysis obtained through equipment installed at small- and medium-sized firm production locations, it is feasible to integrate support with R&D, education, and human resource development commercialization consultancy in the proposed model.

### 4.3. IIoT Block Generation

When IIoT information develops a blockchain from the IIoT device to the end destination in industrial locations, IIoT information generates an arbitrary bit block ${}_{R}{\left\{0,1\right\}}^{n}$ using Equation (1).
(1)$$IIoT\_b_{i}(=h^{i}\left(s\right))(i\in \mathrm{Integer},\;s\in {}_{R}{\left\{0,1\right\}}^{i})$$

As shown in Equation (1), the proposed model duplicates the negotiated IIoT blockchain to an arbitrary block $IIoT\_b_{i}$ size of k bits so that hash processing may be performed without losing IIoT information. The integrity of IIoT data is currently validated by appending signatures to the beginning and end of the hash chain based on the replication’s odd/evenness. This is because a solution such as this can reduce the verification overhead while maintaining the integrity of IIoT data verification.

Equation (2) shows that the proposed model interleaves IIoT unique identification information $IIoT\_I_{i}$ to incorporate IIoT unique identification in the produced bit block (1).
(2)$$\vec{IIoT\_BI_{i}}=IIoT\_b_{i}\;=\;IIoT\_I_{i}$$

Algorithm 1 divides IIoT data into arbitrary chunks and stores it with signature keys and access control restrictions in a blockchain. Moreover, IIoT information and signature keys, computationally demanding data is considered a non-blockchain.
**Algorithm 1.** IIoT block generation.**Input****:** IIoT information generated at the industrial site**Output****:** Create a k-bit blockchain based on odd/even1: Generate IIoT information2: While receive IIoT information3:    if IoT information then4:        Compare IIoT information with IIoT information from each other IIoT sensor5:        Generate and store the IIoT information block6:        While the IIoT information block can be identified7:          if generate random blocks in n-bit form using R{0,1}n from all IIoT information8:          Convert each block to replication9:           if generate hash values for odd/even10:             Add signature to first and last of hash chains11:             Verification of integrity of IIoT information12:             Interleaved process to exclude IIoT bit blocks and IIoT unique identification information 13:           else14:             Regenerate hash values for odd/even15:           end if16:          else17:            Process computationally intensive information into a non-blockchain18:            reconfirm the IoT Informaiton19:          end if20:        do while21:    else22:        Request IoT information 23:    end if24: do while

### 4.4. Information Processing in IIoT

Because IIoT data created at industrial sites is difficult to utilize for business right away, the process of processing, changing, and extracting IIoT data should be performed. The proposed model processes IIoT data in seven phases, as shown in Figure 4.

Step 1: Create IIoT data

This step involves gathering IIoT data from industrial equipment, which requires each piece of equipment to attach or interlock IIoT sensors. IIoT data is divided into three categories: general data, statistical data, and other data, and IIoT data is created in real-time.

Step 2: Deployment of IIoT Networks

In this stage, equipment with IIoT sensors links IIoT information to each other to create a new IIoT network, creating a pair of connections between each IIoT information through link points.

Algorithm 2 depicts a method for constructing a new IIoT network by interconnecting IIoT data. Algorithm 2 displays the freshly produced IIoT data from the IIoT sensor through the link point $P_{i}^{n-1}$. The link point $P_{i}^{n-1}$ is used to construct a connection pair that connects IIoT data.
**Algorithm 2.** IIoT link point build.**Input:** The IIoT’s new connectivity point $P_{i}^{n-1}$ **Output:** IIoT generation of linking pairs1: Creating IIot information links 2: for(*i* = 0; *i* ≤ *n*; *i*++): 3:   for(*j* = 0; *j* ≤ *n* − 1; *j*++): 4:     Create a triple link [$L_{i-1,j-1}$, $L_{i,j}$, $L_{i+1,j+1}$] 5:     Correspondence estimates for the link pairs ($L_{i-1,j-1}$, $L_{i,j}$) and ($L_{i,j}$, $L_{i+1,j+1}$) from two link pairings 6:   end for 7: end for

($L_{i-1,j-1}$, $L_{i,j}$) and ($L_{i,j}$, $L_{i+1,j+1}$) is the link pair of IIoT information. It is formed in the form of ($L_{i-1,j-1}$, $L_{i,j}$) and ($L_{i,j}$, $L_{i+1,j+1}$). The function of these pairs of connections is to increase the integrity verification and management efficiency IIoT information by linking IIoT information together.

Step 3: IIoT data transmission to IIoT servers

The IIoT network environment maintains a network speed above LTE level and a network environment to reduce network latency throughout this stage of delivering IIoT information to the IIoT server.

Step 4: Information from the IIoT is pre-processed

This step involves pre-processing structured and unstructured data from IIoT sensors, which is performed using various methods to collect information needed by industrial sites in real-time. In this case, hash chains are used to aggregate, and process n pieces of IIoT data, and the technique utilized is presented in Algorithm 3. The IIoT information in Algorithm 3 is completely connected by repeating the number of IIoT sensors of equipment placed at industrial locations, resulting in a connection offset.
**Algorithm 3.** Gathering algorithm of IIoT.**Input:** Information gathered from the IIoT **Output:** Calculate the offset of IIoT connection information1: Verify IIoT connection details 2: Set the number of IIoT connection parameters: 0 for initial value [0] 3: Set the IIoT connection index to zero 4: for(*i* = 0; *i* ≥ *n*; *i*++) 5:  index = { *i* | GI ∈ Z, Z is integer} 6:  Intitial_value[index] ± Initial_value[index] 7:  offset [index] = Intitial_value[index] + linking_length 8:  index ++ 9: end for

Step 5: IIoT information and blockchain setup

This step involves creating a blockchain by adding additional IIoT information $I_{i,j}$ to connection information <$I_{i,j}^{n-1}$,$\;P_{i,j}^{n-1}$ > ∈ $R_{i,j}^{n-1}$. IIoT information is generally exchanged through a connection offset of IIoT information, with a thorough operation methodology as shown in Algorithm 4. Algorithm 4 adds an offset to the IIoT information acquired from the IIoT sensor based on the blockchain, updating the association point $P_{i,j}^{n-1}$ of the newly added IIoT information <$I_{i,j}^{n-1}$,$\;P_{i,j}^{n-1}$ >. The rationale is that IIoT data may be updated while minimizing the cost of configuring blockchain between IIoT data.
**Algorithm 4.** IIoT information and blockchain setup.**Input:** The IIoT connectivity information utilizing offset **Output:** Using connection points $P_{i,j}^{n-1}$, update IIoT data jointly (*n* − 1)1: Gather IIoT connectivity information 2: for *i* from 1 to *n*: 3:  for each piece of linking information 4:    To produce new IIoT connecting information, add offset[i] to each IIoT information 5.    Refresh the link information $P_{i,j}^{n-1}$ of IIoT at linking information <$L_{i,j}^{n-1}$, $P_{i,j}^{n-1}$> ∈ $R_{i,j}^{n-1}$ 6:    IoT data configuration on the blockchain 7:  end for 8: end for

Step 6: Information Analysis and Monitoring in the IIoT

This step involves using data mining techniques (statistics, text mining, visualization, etc.) to study and monitor data processed by the IIoT server, and IIoT information is processed, processed, assessed, and analyzed. In this case, the manager is responsible for both monitoring and managing the analysis outcome.

Step 7: Management and usage of services

This step uses and manages services based on analysis findings so that IIoT data may be utilized in the industry based on monitoring data and to support services by analyzing machine learning/R-based statistical data.

### 4.5. IIoT Information Reinforcement Learning

The index, collection location (binary value of X-axis and Y-axis coordinates), group index, type, and type of IIoT information are considered in the proposed model for reinforcement learning of IIoT details, as shown in Figure 5. According to the aim and features of IIoT, the suggested model creates and learns IIoT information using a machine learning-based method, as illustrated in Figure 5.

As shown in Figure 5, the proposed model load balances IIoT information so that it may be asymmetrically connected, then orthogonalizes IIoT information to group IIoT information asymmetrically using the weight probability function IIoT information, as indicated in Equation (3). This is because IIoT details may be secured outside using hash processing. Interference between IIoT data was reduced by classifying it according to its kind and features and producing it regularly.
(3)$$-\sum_{i=1}^{n}\frac{1}{n}\log\frac{1}{n}=\log\;n$$
where *i* is the number of IIoT data associations.

The proposed model diversifies the IIoT data collecting model by processing the dimensions of vectorized IIoT data in multiple ways, as shown in Figure 6. In identifying missing values of IIoT information, the dropout ratio in Figure 6 is utilized as a restriction to prevent the overfitting of IIoT details. To strengthen the expressive power of the IIoT information gathering model, the Dense is raised by $2^{n}$ times to reduce overfitting.

Table 3 shows how the proposed model extracts IIoT data using probabilistic statistical techniques. Table 3 shows how to build up a layer (L) using a machine learning deep learning model, set input and output data, and learn to obtain an extraction value. The proposed model enhances computational efficiency by reducing dimension d to feature subspace (k), which is assisted in the proposed model by leveraging the importance of security information in bit form while lowering the burden on IIoT information. IIoT data is handled asymmetrically if the signature value of IIoT information is randomly chosen to link IIoT information with the hash chain.

## 5. Evaluation

### 5.1. Environment Setting

The proposed model’s experimental setting was setup as stated in Table 4, and the simulation was run. The simulation environment used an OPNet simulator to simulate the production site, and the obtained data was combined with information gathered on the production site. Google Collaboration provided programming languages and frameworks for analyzing IIoT datasets.

The network has a range of 100 m and a rental bandwidth of 5 MHz/2.5 MHz. The typical transaction size was set to 100–200 B, with a variance compensation coefficient limit time of 0.002 s/KB. Table 4 shows the additional options.

### 5.2. Configuring the Model

The proposed model divided IIoT datasets into training and test sets using classification (LR, Multilayer Perceptron (MLP)), Support Vector Machines (SVM), and learning (K-nearest neighbors (KNNs)), and Random Forest (RF) methods. Table 5 shows the parameter values for each method utilized in the proposed model. The model was built with five specified features (index, collection location (binary value of X-axis and Y-axis coordinates), group index, type, etc.) for machine learning.

### 5.3. Performance Evaluation

#### 5.3.1. Performance Evaluation by Category

IIoT information was segregated into blockchain and non-blockchain-based scenarios to verify the integrity of IIoT information provided by IIoT sensors at production sites, and efficiency, throughput, overhead, and other factors were assessed in Table 6 based on the number of simulated routers. Because the proposed model uses blockchain to handle IIoT data, the average performance efficiency was 16.9%, the average processing rate was 12.1%, and the average overhead was 21.1% lower than the present environment without a blockchain-based IIoT. Because a blockchain-based IIoT sensor is applied to the line that creates IIoT information at the manufacturing facility, no extra function is required for the production process. The time required to verify the integrity of IIoT data is reduced. The proposed model, in particular, was able to achieve better outcomes because it could evaluate IIoT device groups separately for administrators to assess by processing IIoT data that may occur at endpoints at production sites as blockchain.

#### 5.3.2. Reinforcement Algorithm Evaluation

The performance analysis findings for each algorithm employed in the proposed model are shown in Table 7. As shown in Table 7, the proposed model evaluated data obtained from IIoT sensing using an algorithm based on the number of simulated clouds. The accuracy changed by 2.5% to 3% as the IIoT information grew. When comparing performance before and after dimension reduction, Multilayer Perceptron (MLP) performed well before dimension reduction, whereas Random Forest (RF) performed well. Regarding IIoT data collecting performance, the smaller the data collection, the better the Support Vector Machine (SVM) method, and the bigger the data collection, the better the Multilayer Perceptron algorithm (MLP).

#### 5.3.3. Comparison of Time Complexity

Server computing complicated, IIoT computing complex, verifier computing complex, verifier storage complex, communication, complexity blockchain generation complex, and Table 8 were used to classify the time complexity between the proposed and current models. Because IIoT request and answer messages are gathered across all servers in an IIoT ecosystem, IoT edge servers cryptographically enhance data or decrease communication before IIoT ensures the integrity of IIoT information delivered to the server. Meta-information must be saved in advance to validate the IIoT information stored in the server. Because the validated data is so extensive, the amount of this meta-information should be kept as little as is feasible. In this situation, the IIoT must store more data while keeping the IIoT data to a bare minimum. Because the verifier handles the information on the outside without extra processing, the proposed model reduces the amount of computing required to process the integrity of IIoT details compared with the present paradigm. Verifier complexity of O(logn) was demonstrated since the suggested model hierarchically splits IIoT information into subgroups and combines IIoT’s critical data. Because it verifies the integrity of IIoT information by replicating the negotiated blockchain of IIoT information to any block size of kbits without loss of IIoT information by adding signatures at the beginning and end of the hash chain according to odd/even of the replication, the proposed model has the same time complexity as O(slogn).

#### 5.3.4. Discussions

The pace at which IIoT data is collected, processed, and interpreted must be enhanced to efficiently manage the production and delivery of physical commodities in the manufacturing business; it is much less likely that data will be intercepted or violated. Large corporations are improving their competitiveness by investing in IIoT and future technologies (digital twin, electronic recorder (ELD), intelligent edge, predictive maintenance, and RFID) in the manufacturing business (Radio Frequency Identification). However, the cost is insufficient in small-scale manufacturing sectors, such as small- and medium-sized firms, and the latest technologies are not adopted into the manufacturing industry. The proposed model is based on an IIoT information platform that uses a blockchain to advance IIoT information while lowering costs in a small manufacturing business. The proposed model is the most crucial aim of IIoT information progress, and it can be used to improve the integrity of IIoT data and connect disparate IIoT data. The proposed model was tested in contexts where vast quantities of varied items are produced in tiny production facilities. Compared with the prior environment without a blockchain-based IIoT information platform, efficiency, throughput, and overhead were all improved. However, because the suggested model is tailored for small manufacturing industries, optimal performance is unlikely in big manufacturing industries. Large-scale manufacturing industry locations must be addressed differently from small-scale industry settings. Therefore, extra needs must be reflected. Furthermore, future technologies (digital twin, electronic recorder (ELD), intelligent edge, predictive maintenance, and Radio Frequency Identification (RFID)) are used in various combinations at the manufacturing site. Different results are obtained depending on the manufacturing industry’s conditions.

## 6. Conclusions

Industrial sites are making a lot of efforts to reduce production costs by improving the efficiency and integrity of IIoT sensors. In most industrial sites, there is an urgent need to ensure the integrity of IIoT information because there is a lot of information that is abnormally generated from IIoT sensors. In this study, we proposed an IIoT information reinforcement model based on the IIoT information platform using blockchain to ensure the integrity of IIoT information generated at industrial sites. The proposed model senses IIoT information in real-time regardless of time and place and performs close linkage between IIoT information with different characteristics. The proposed model constructed a subnet by probabilistic use of the attribute frequency of IIoT information in order to efficiently collect and analyze IIoT information. The deployed subnets were grouped by small blockchain to improve dynamic clustering speed and accuracy, and then gradually ensured the integrity of IIoT information. The performance evaluation for the proposed blockchain-based IIoT solution shows that its efficiency is 16.9% better, its average processing rate is 12.1% faster, and its average overhead is 21.1% lower than for the present environment without blockchain-based IIoT. In addition, the proposed model analyzed the data collected from IIoT sensing by algorithm according to the number of simulated clouds and found that the accuracy differed by 2.5% to 3% on average as the IIoT information increased constantly. In future studies, based on the results of this study additional research and supplementation will be conducted for IIoT sensors attached to other production facilities in industrial sites.

## Figures and Tables

**Figure 1 sensors-22-04645-f001:**
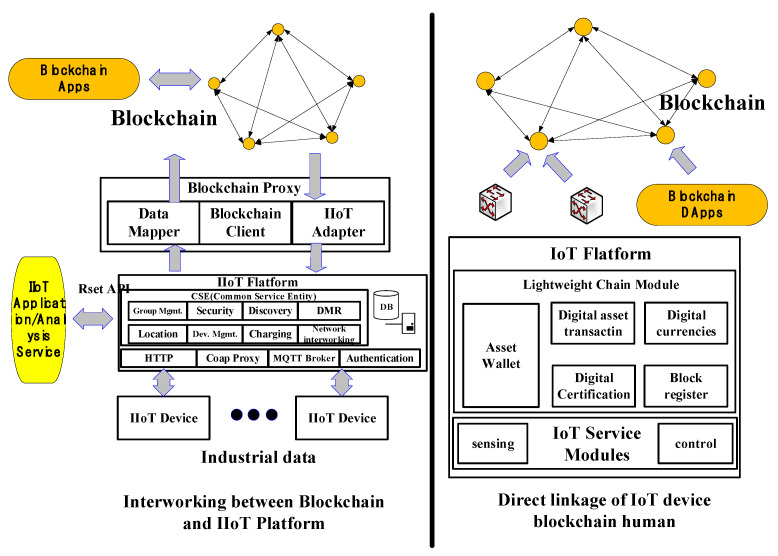
Interaction between blockchain and IoT for production sites.

**Figure 2 sensors-22-04645-f002:**
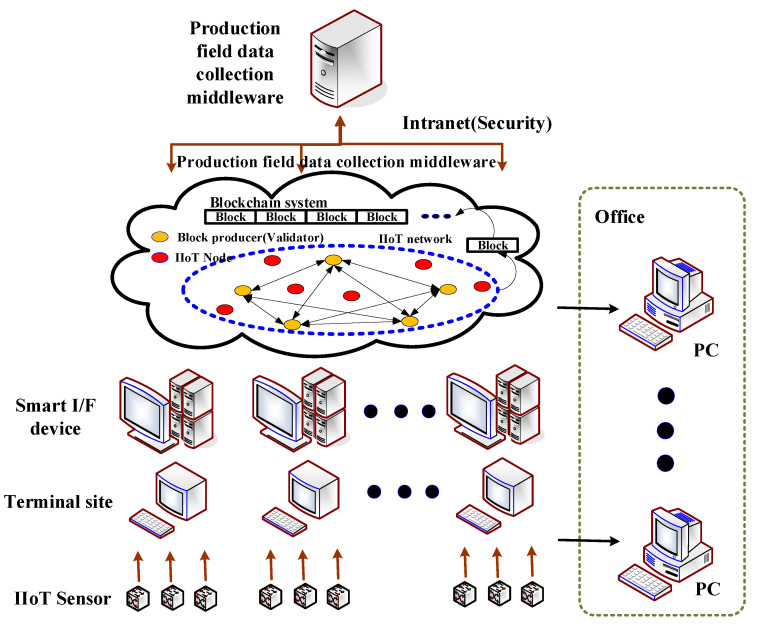
IIoT information gathering process of proposed model.

**Figure 3 sensors-22-04645-f003:**
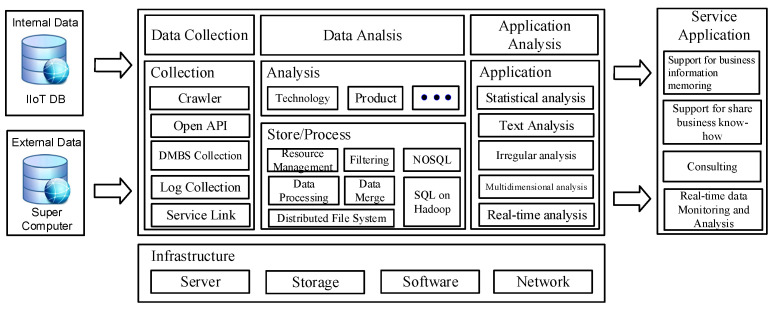
Gathering and sharing of information for platform capabilities.

**Figure 4 sensors-22-04645-f004:**
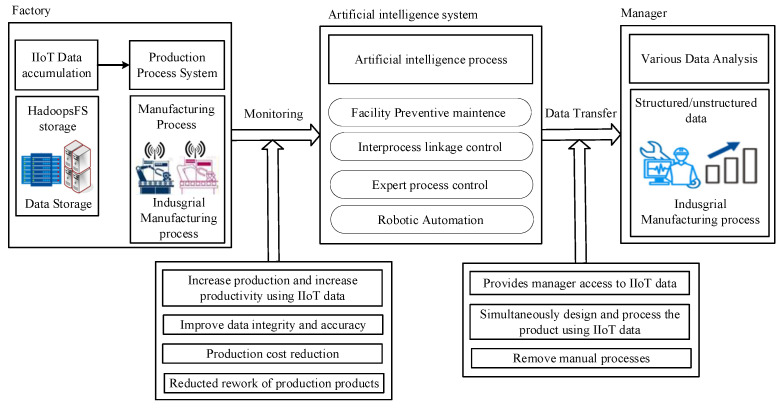
Processing of the proposed model.

**Figure 5 sensors-22-04645-f005:**
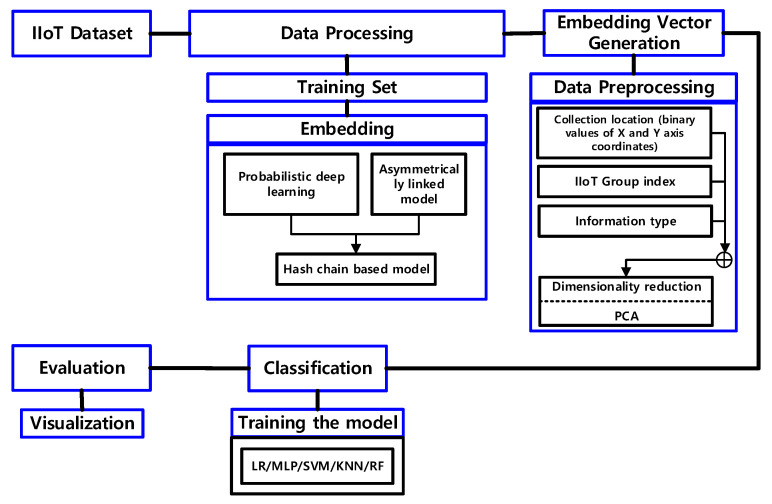
Processing of the proposed model.

**Figure 6 sensors-22-04645-f006:**
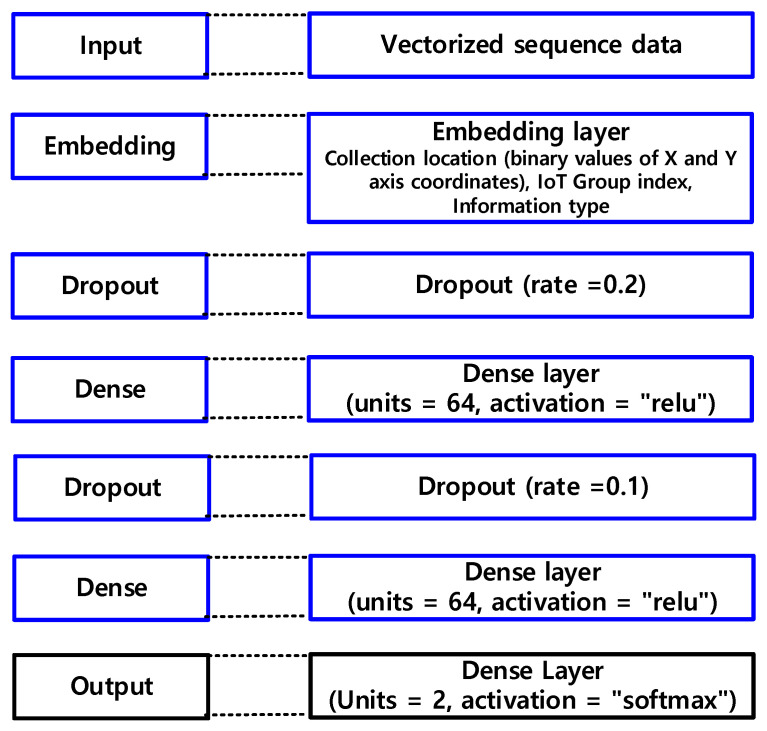
Detection of missing values of IIoT information.

**Table 1 sensors-22-04645-t001:** Smart factory vs. factory automation.

Division	Smart Factory	Factory Automation
Definition	An intelligent automation platform that enhances quality and performance by automating environmental safety, marketing, design, process, and shipping at the plant.	A production system mechanism that manages the functioning of a machine without the need of people.
Advantage	Scripted languages and compiler languages such as C++ are used to create available software components.	For automation and digitalization, integrate internet of things (IoT), Artificial Intelligence (AI), Big Data, and other technologies.
Weakness	Make a substantial first investment.	Connection, collection, and analysis of data created during manufacturing are difficult.
Object	Manufacturing’s procurement, logistics, and consumer.	Computers and robots are examples of equipment.
Function	Automate unmanned and manufacturing operations across the plant utilizing computers and robots.	Provide each thing intelligence and link it to the internet of things to connect, gather, and analyze data autonomously (IoT).
Range	‘Smart’ horizontal integration.	Vertical integration, ‘manualizing’ and ‘factory’.
Characteristics	Combine the newest technology to boost efficiency and productivity across management, from industrial operations to management.	Integrate corporate processes with an emphasis on production management.

**Table 2 sensors-22-04645-t002:** Analysis of related works.

Related Works	Related Technique	Advantages	Limitations
Sharing cloud data [20,23]	Bitcoin, incentives for data sharing are highly valued.	Real-time data storage, ensure data integrity and information security, cloud data sharing should be rewarded fairly.	Collaboration between IIoT devices is required, blockchain is used to store shared data.
Data gathering and sharing in the IIoT [24]	Deep reinforcement learning and mobile crowdsensing.	Collaborative data collection and sharing.	Blockchain is used to store shared data.
secureSVM: Data exchange in smart cities [17,22,25]	Decision Making Trial and Evaluation Laboratory (DEMATEL) Support vector machine (SVM), Paillier, gradient descent.	Describe the many concerns and obstacles associated with IoT and blockchain implementation, IoT data transmission with privacy protection for SVM training.	Proactive government and other policies, blockchain is used to store shared data.
Storage systems that share data [19,26]	Decisional Bilinear Diffie–Hellman (DBDH), attributed-based encryption, interplanetary file system (IPFS).	Integrate identity verification, and at any time, you may trace and terminate malicious users, decentralized storage and fine-grained access control for shared data.	Ensuring Decisional Bilinear Diffie–Hellman (DBDH) for system security, IPFS storage systems have no incentives.
Sharing of electronic health records [27]	IPFS, smart contract.	Dependable access control, decentralized storage.	IPFS storage systems have no incentives.
Sharing of electronic medical records [28]	IPFS, encryption system with attributes.	Data access control and decentralized storage without affecting retrieval.	IPFS storage systems have no incentives.
IoT Platform and System [15,16,18]	Supply chain, autonomous vehicle solutions, manufacturing plant asset management.	New commercial prospects, regulatory restrictions, and measures to promote transparency and visibility are discussed.	Blockchain is used to store shared data.
Smart Grid System [21]	P2P network, data load balancing.	Model efficacy is evaluated using energy management costs and transmission delay characteristics.	Assume P2P network is secure.

**Table 3 sensors-22-04645-t003:** Features extract and selection equation of IIoT data.

Feature	Equation
Mean	$\bar{X}$= $\frac{1}{n}\overset{N}{\underset{i=1}{\sum }}X_{i}$
Variance	$\sigma^{2}$= $\frac{1}{n}\overset{N}{\underset{n=1}{\sum }}{\left(X_{i}-\bar{X}\right)}^{2}$
Maximum	$X_{max}$= $\begin{array}{c} max\\ i=1,2,\ldots ,N \end{array}\left(X_{i}\right)$
Minimum	$X_{min}$= $\begin{array}{c} min\\ i=1,2,\ldots ,N \end{array}\left(X_{i}\right)$
Root mean square	S= $\sqrt{\frac{1}{n}\overset{N}{\underset{i=L}{\sum }}X_{i}^{2}}$

**Table 4 sensors-22-04645-t004:** Simulation environment.

Division	Specification	
OS	OPNet simulation	
VMware Workstation	VMware Workstation 14	
	OS	Windows 10 Professional
	CPU	1 core
	RAM/HDD	8 GB/150 GB
Language	Python 3.9.5	
Tool	Google Colab	
Library	Scikit-learn, matplot, numpy, pandas, etc.	
The transmit/receive power of the IIoT	0.1 W/0.15 W	
The network coverage radius	100 m	
The static circuit power	0.01 W	
The pathloss exponent	1	
The subnet tree depth	4	
$\mathrm{The}\;\mathrm{available}\;\mathrm{bandwidth}\;\mathrm{for}\;\beta_{Server}$ $/\beta_{IIoT}$	5 MHz/2.5 MHz	
The power of noise	−174 dBm/Hz	
Subnet storage capacity	0.25 TB	
Input data size	3 kbits/s	
Delay threshold	8s	
Link capacity	5 Gbps	
Poisson lambda	85%	
Data generation span	5 min	
Max access count	20	
The unit price of energy	0.2 Token/J	

**Table 5 sensors-22-04645-t005:** Algorithm parameter values.

Algorithm	Parameter Value
LR	Regularization strength(C): 5, 10, 15; solver: lbfgs; penalty: l2
MLP	Hidden_layer: 50; activation: relu; weight optimization: adam; learning rate: 0.001
SVM	Regularization parameter(C): 0.001, 0.01, 1, 10, 25, 50; kernel: rbf; probability: true
KNN	Number of neighbors: 3∼5; weights: uniform
RF	number of trees: 50

**Table 6 sensors-22-04645-t006:** Performance category assessment.

Category		The Count Number of Simulations					
		1	2	3	4	5	6
Efficiency (%)	Not using blockchain	24.2	23.3	22.9	25.1	27.3	26.5
	Using blockchain	33.7	40.2	39.2	36.8	38.7	35.6
Throughput (%)	Not using blockchain	1.21	1.44	1.33	1.47	1.29	1.39
	Using blockchain	2.21	2.11	1.99	1.83	2.06	2.14
Overhead (%)	Not using blockchain	16.73	20.31	19.26	21.05	18.97	17.47
	Using blockchain	9.32	10.04	9.98	12.65	11.65	13.11

**Table 7 sensors-22-04645-t007:** Evaluation by reinforcement algorithm.

**LR**	CN	1	2	3	4	5	6
	A	79.15	82.08	81.74	83.77	82.17	84.08
	LT	0.006	0.007	0.006	0.007	0.006	0.009
	F1	80.65	82.14	81.08	82.65	84.06	83.38
**MLP**	CN	1	2	3	4	5	6
	A	89.07	90.95	91.39	91.05	90.11	92.36
	LT	0.004	0.005	0.003	0.006	0.007	0.005
	F1	91.30	89.47	87.98	90.43	88.12	90.61
**SVM**	CN	1	2	3	4	5	6
	A	84.71	86.37	85.02	89.01	86.98	85.49
	LT	0.006	0.007	0.006	0.007	0.006	0.006
	F1	84.12	86.08	85.99	87.15	86.07	85.35
**KNN**	CN	1	2	3	4	5	6
	A	86.37	84.92	86.71	87.05	85.63	87.41
	LT	0.003	0.004	0.005	0.004	0.006	0.005
	F1	83.03	86.32	82.98	85.02	83.84	85.69
**RF**	CN	1	2	3	4	5	6
	A	88.27	89.42	88.78	90.14	89.89	90.36
	LT	0.004	0.003	0.003	0.005	0.004	0.006
	F1	87.09	85.47	88.14	90.41	87.38	86.25

CN: the count number of simulations; A: accuracy; LT: learning time; F1: F1-score.

**Table 8 sensors-22-04645-t008:** Comparisons of time complexity.

Model	IIoT Edge Server Computation Complexity	IIoT Computation Complexity	Verifier Computation Complexity	Verifier Storage Complexity	Communication Complexity	Blockchain Generation Complexity
[20,23]	O(logn)	O(1)	O(${\mathrm{logn}}^{2}$)	O(logn)	O(1)	O(${\mathrm{logn}}^{2}$)
[24]	O(logn)	O(logn)	O(nlogn)	O(1)	O(nlogn)	O(logn)
[17,22,25]	O(1)	O(logn)	O(${\mathrm{logn}}^{2}$)	O(nlogn)	O(1)	O(slogn)
[19,26]	O(nlogn)	O(logn)	O(1)	O(1)	O(clogn)	O(nlogn)
[27]	O(1)	O(logn)	O(logn)	O(nlogn)	O(${\mathrm{logn}}^{2}$)	O(logn)
[28]	O(logn)	O(1)	O(logn)	O(nlogn)	O(${\mathrm{logn}}^{2}$)	O(nlogn)
[15,16,18]	O(nlogn)	O(1)	O(1)	O(1)	O(1)	O(1)
Proposed model	O(logn)	O(logn)	O(logn)	O(logn)	O(clogn)	O(slogn)

n: total number of message blocks, c: number of sampling messages, s: number of block signatures in the network.

## Data Availability

Publicly available datasets were analyzed in this study. The data presented in this study are available on request from the corresponding author.

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
