# Peer review of "Secure IIoT Information Reinforcement Model Based on IIoT Information Platform Using Blockchain"

_sensors, 2022, doi:10.3390/s22124645_

Round 1

Reviewer 1 Report

Need spell check required

Author Response

As reviewer request, the grammatical mistakes is modify as follows:

-. This manuscript had been modified from native english writer.

-. This manuscript had been checked spell check and grammar using grammarly.com.

Reviewer 2 Report

The manuscript proposes a blockchain-based IIoT information platform. The topic is of current interest, and the manuscript is well organized. However, we have the following concerns:

- Blockchain-based IIoT has been studied extensively. Compared to existing blockchain-based IIoT platforms, what is the novelty of this manuscript?

- The main contribution should be summarized at the end of the introduction.

- The authors are supposed to give a problem definition right after the system model.

- The performance evaluation is limited. For example, the experiment settings are not described.

Author Response

- Blockchain-based IIoT has been studied extensively. Compared to existing blockchain-based IIoT platforms, what is the novelty of this manuscript?

* Response of Reviewer

As reviewer request, Compared to the existing blockchain-based IIoT platform, the proposed IIoT platform has two major innovations. First, the proposed model enables the enhancement of the integrity of IIoT information and close linkage of different IIoT information to enhance IIoT information while minimizing costs in small manufacturing industries. Second, in order to accurately classify large amounts of IIoT sensing information, information sharing and system linkage within industrial sites are possible through a probability-based deep learning clustering process. Related contents were included in the summary, section 1, section 4, and conclusion.

- The main contribution should be summarized at the end of the introduction.

* Response of Reviewer

As reviewer request, The main contribution was summarized and added at the end of the introduction.

- The authors are supposed to give a problem definition right after the system model.

* Response of Reviewer

As reviewer request, Problem definitions are described in Section “5.3.3 Discussion”.

- The performance evaluation is limited. For example, the experiment settings are not described.

* Response of Reviewer

As reviewer request, Experimental items related to performance evaluation were added to Table 6.

Reviewer 3 Report

In this paper, the authors present a blockchain-based IIoT information reinforcement model. The proposed model processes and transmits information generated an IIoT device to the IIoT server independently.

The paper does not present noticeable contribution. Many similar solutions were presented in recent research, and the authors failed to establish the superiority of their proposed work over other works.

Major Comments:
1. the introduction does not provide a proper context to the problem being addressed by this research. There needs to be explanation on what the problem is and why the proposed system is needed.

2. The related works section does not present the related works properly. The first two sub sections should be moved out of related works to be "Background" section. Subsection 2.3 should be th "Related Works" section. However , it does not present any meaningful summaries of previous works presented to address the problem. Many other works such as:
https://ieeexplore.ieee.org/abstract/document/9314268
https://ieeexplore.ieee.org/abstract/document/8780161
https://ieeexplore.ieee.org/abstract/document/9165552
https://www.researchgate.net/profile/Vinod-Kumar-5/publication/344712761_Integration_of_Next_Generation_IIoT_with_Blockchain_for_the_Development_of_Smart_Industries/links/5f900efa299bf1b53e37a95c/Integration-of-Next-Generation-IIoT-with-Blockchain-for-the-Development-of-Smart-Industries.pdf

The works listed above are just a simple example of many other works addressing the same problem. The authors need to discuss these other works so they can present the case that their proposed work is better.

3. Section 3 presents the blockchain solution very briefly without proper explanations. Further details are needed about the blockchain implementation because it is the core of the paper's contribution.

4. The paper does not present a "Discussions" section. There needs to be a detailed discussion section where the results are deeply discussed with meaningful insights into what these results mean.

5. Due to the lack of a discussions section, the proposed system is not compared to any other related works at all. Not in terms of features, and not in terms of performance. There is no way for the reader to understand the value of the proposed system.

Minor Comments:
1. The paper could definitely use some proofreading to improve its readability and flow.
2. Avoid the use of acronyms in the abstract.

Author Response

Major Comments:

  1. the introduction does not provide a proper context to the problem being addressed by this research. There needs to be explanation on what the problem is and why the proposed system is needed.

* Response of Reviewer

As reviewer request, the "1. Introduction" section adds an explanation of what problems are given and why the proposed model is needed.

  1. The related works section does not present the related works properly. The first two sub sections should be moved out of related works to be "Background" section. Subsection 2.3 should be th "Related Works" section. However , it does not present any meaningful summaries of previous works presented to address the problem. Many other works such as:

https://ieeexplore.ieee.org/abstract/document/9314268

https://ieeexplore.ieee.org/abstract/document/8780161

https://ieeexplore.ieee.org/abstract/document/9165552

https://www.researchgate.net/profile/Vinod-Kumar-5/publication/344712761_Integration_of_Next_Generation_IIoT_with_Blockchain_for_the_Development_of_Smart_Industries/links/5f900efa299bf1b53e37a95c/Integration-of-Next-Generation-IIoT-with-Blockchain-for-the-Development-of-Smart-Industries.pdf

The works listed above are just a simple example of many other works addressing the same problem. The authors need to discuss these other works so they can present the case that their proposed work is better.

* Response of Reviewer

As reviewer request, the first two subsections of Section 2 were rewritten as "2. Background" sections. Subsection 2.3 consists of a "3. Related Studies" session. "3. Related Research" adds content on topics such as the proposal model, and summarizes the characteristics of each topic in Table 2.

  1. Section 3 presents the blockchain solution very briefly without proper explanations. Further details are needed about the blockchain implementation because it is the core of the paper's contribution.

* Response of Reviewer

As reviewer request, added the section "4.3 Create Blocks" to implement the proposed model's blockchain.

  1. The paper does not present a "Discussions" section. There needs to be a detailed discussion section where the results are deeply discussed with meaningful insights into what these results mean.

* Response of Reviewer

As reviewer request, we added section "5.3.4 Discussion".

  1. Due to the lack of a discussions section, the proposed system is not compared to any other related works at all. Not in terms of features, and not in terms of performance. There is no way for the reader to understand the value of the proposed system.

* Response of Reviewer

As reviewer request, the "5.3.4 Discussion" section describes the problems in previous studies, the functions and characteristics of the proposed model, and the solutions in future studies.

Minor Comments:

  1. The paper could definitely use some proofreading to improve its readability and flow.

* Response of Reviewer

As reviewer request, the paper was calibrated to improve the readability of the paper.

  1. Avoid the use of acronyms in the abstract.

* Response of Reviewer

As reviewer request, we modified it to avoid using acronyms in the summary.

Reviewer 4 Report

This paper proposed an approach to processes and transmits information  generated by the same IIoT device to the IIoT server independently, allowing for blockchain processing of IIoT data that may occur at endpoints in industrial sites. This also creates a hierarchical subnet using probabilistic reflection of IIoT data on attribute value frequency. This paper is well written and the contributions are good for a journal article. However, there are certain changes required before consider this paper for publication. Follow the comments:

1. The literature can be consider the recently published relevant articles.

2. The section heading are too long and not relevant. It is recommended to rewrite the precise section and subsection headings.

3. The Theoretical analysis of the proposed work is poor. It is recommended to provide the theoretical discussion on proposed work.

4. Determine the time complexity of the proposed algorithms and also compare it using the existing models.

5. The discussion on reinforcement learning are poor. It is recommended to present clearly, which are the states and actions in the proposed work. It is also not clear about the reward function in this paper.

6. Provide the reasons for achieving the superior performance of the proposed work over the existing ones.

7. The abstract and conclusion are avoided generic discussions, and provide a precise contributions and their pitfalls along to a summary on results.

Author Response

  1. The literature can be consider the recently published relevant articles.

* Response of Reviewer

As reviewer request, Recently, related studies were added to the "3. Related Studies" section, and the related technologies, advantages, and disadvantages of the existing research are summarized in Table 2.

  1. The section heading are too long and not relevant. It is recommended to rewrite the precise section and subsection headings.

* Response of Reviewer

As reviewer request, The title of the sections and subsections of the thesis composition has been implicitly rewritten.

  1. The Theoretical analysis of the proposed work is poor. It is recommended to provide the theoretical discussion on proposed work.

* Response of Reviewer

As reviewer request, The "5.3.4 Discussion" section additionally describes the problems of the existing research, the functions and characteristics of the proposed model, and the excellent performance.

  1. Determine the time complexity of the proposed algorithms and also compare it using the existing models.

* Response of Reviewer

As reviewer request, the results of comparing the time complexity of the proposed algorithm with the existing model are added to Section 5.3.3 and Table 10.

  1. The discussion on reinforcement learning are poor. It is recommended to present clearly, which are the states and actions in the proposed work. It is also not clear about the reward function in this paper.

* Response of Reviewer

As reviewer request, The following contents have been modified and added.
- Contents related to reinforcement learning were added to section 4.5.
-. Experimental items related to performance evaluation were added to Table 6.
-. The compensation function of the proposed model was added to the "5.3.4 Discussion", etc.

  1. Provide the reasons for achieving the superior performance of the proposed work over the existing ones.

* Response of Reviewer

As reviewer request, The "5.3.4 Discussion" section additionally describes the problems of the existing research, the functions and characteristics of the proposed model, and the excellent performance.

  1. The abstract and conclusion are avoided generic discussions, and provide a precise contributions and their pitfalls along to a summary on results.

* Response of Reviewer

As reviewer request, In the "1. Introduction" and "6. Conclusion" sections, we added an explanation of what the problem was given, why the proposed model was needed, and further explained the contribution of the proposed model.

Round 2

Reviewer 3 Report

I would like to thank the authors for addressing my previous comments. I have not further comments.

Reviewer 4 Report

None